# Antibiotic Resistance and Mobile Genetic Elements in Extensively Drug-Resistant *Klebsiella pneumoniae* Sequence Type 147 Recovered from Germany

**DOI:** 10.3390/antibiotics9100675

**Published:** 2020-10-05

**Authors:** Kyriaki Xanthopoulou, Alessandra Carattoli, Julia Wille, Lena M. Biehl, Holger Rohde, Fedja Farowski, Oleg Krut, Laura Villa, Claudia Feudi, Harald Seifert, Paul G Higgins

**Affiliations:** 1Institute for Medical Microbiology, Immunology and Hygiene, University of Cologne, 50935 Cologne, Germany; kyriaki.xanthopoulou@uk-koeln.de (K.X.); julia.wille_@uk-koeln.de (J.W.); harald.seifert@uni-koeln.de (H.S.); 2German Centre for Infection Research (DZIF), Partner site Bonn-Cologne, 50935 Cologne, Germany; lena.biehl@uk-koeln.de (L.M.B.); fedja.farowski@uk-koeln.de (F.F.); 3Department of Molecular Medicine, Sapienza University of Rome, 00185 Rome, Italy; alessandra.carattoli@uniroma1.it; 4Department I of Internal Medicine, Faculty of Medicine and University Hospital of Cologne, University of Cologne, 50937 Cologne, Germany; 5Institute for Medical Microbiology, Virology and Hygiene, University Medical Centre Hamburg-Eppendorf, 20246 Hamburg, Germany; rohde@uke.de; 6German Centre for Infection Research, Partner site Hamburg-Lübeck-Borstel, 20246 Hamburg, Germany; 7Department of Internal Medicine II, Infectious Diseases, University Hospital Frankfurt, Goethe University Frankfurt, 60590 Frankfurt am Main, Germany; 8Paul-Ehrlich Institute, Federal Institute for Vaccines and Biomedicine, 63225 Langen, Germany; oleg.krut@pei.de; 9Department of Infectious Diseases, Istituto Superiore di Sanità, 00161 Rome, Italy; laura.villa@iss.it; 10Institute of Microbiology and Epizootics, Centre for Infection Medicine, Department of Veterinary Medicine, Freie Universität Berlin, 14163 Berlin, Germany; claudia.feudi@fu-berlin.de

**Keywords:** carbapenem resistance, carbapenemase, whole genome sequencing, long reads, plasmid, *Klebsiella pneumoniae*, extensively drug-resistant, molecular typing

## Abstract

Mobile genetic elements (MGEs), especially multidrug-resistance plasmids, are major vehicles for the dissemination of antimicrobial resistance determinants. Herein, we analyse the MGEs in three extensively drug-resistant (XDR) *Klebsiella pneumoniae* isolates from Germany. Whole genome sequencing (WGS) is performed using Illumina and MinION platforms followed by core-genome multi-locus sequence typing (MLST). The plasmid content is analysed by conjugation, S1-pulsed-field gel electrophoresis (S1-PFGE) and Southern blot experiments. The *K. pneumoniae* isolates belong to the international high-risk clone ST147 and form a cluster of closely related isolates. They harbour the *bla*_OXA-181_ carbapenemase on a ColKP3 plasmid, and 12 antibiotic resistance determinants on an multidrug-resistant (MDR) IncR plasmid with a recombinogenic nature and encoding a large number of insertion elements. The IncR plasmids within the three isolates share a high degree of homology, but present also genetic variations, such as inversion or deletion of genetic regions in close proximity to MGEs. In addition, six plasmids not harbouring any antibiotic resistance determinants are present in each isolate. Our study indicates that genetic variations can be observed within a cluster of closely related isolates, due to the dynamic nature of MGEs. The mobilome of the *K. pneumoniae* isolates combined with the emergence of the XDR ST147 high-risk clone have the potential to become a major challenge for global healthcare.

## 1. Introduction

The evolution and spread of antibiotic-resistant pathogens has emerged as one of the most important public health problems worldwide over the last decades (https://www.who.int/en/news-room/fact-sheets/detail/antibiotic-resistance). In bacterial genomes, capture, accumulation and dissemination of antibiotic resistance determinants are often associated with mobile genetic elements (MGEs) like plasmids, transposons and insertion sequences (ISs) [1]. Plasmids are often assemblies of different MGE modules and are the most efficient intra- and interspecies DNA transfer mechanism among prokaryotes [2]. This is well exemplified by the global spread of the KPC carbapenemase involving the incompatibility group FIIk (IncFIIk) plasmids in *Klebsiella pneumoniae* [3]. Moreover, *b*la**_NDM-1_ in *K. pneumoniae* has been mainly associated with broad host range IncA/C2, IncHI1, IncX3 and IncN2 plasmids [4]. In *Acinetobacter baumannii,* the transposon Tn*125,* harbouring the insertion element IS*Aba125,* is considered as the main vehicle for the dissemination of NDM-1 enzymes [5,6].

*K. pneumoniae*, belonging to the Enterobacterales family, is a natural inhabitant of the gastrointestinal tract of humans and animals. Nevertheless, it is also encountered as a nosocomial pathogen causing various infections such as pneumonia, urinary tract infection and bloodstream infection [4]. Of concern is the rapid expansion of carbapenem-resistant *K. pneumoniae*, mainly associated with those carbapenemases which are endemic in certain countries, such as KPC-positive *K. pneumoniae* in Greece and Italy [7,8]. OXA-48-like is the most common carbapenemase in Enterobacterales in some regions of the world including Germany. Other frequently encountered carbapenemases in Germany include VIM-1 and NDM-1 [9,10]. The successful propagation of OXA-48-positive Enterobacterales is reinforced by the global distribution of certain high-risk clones (e.g., *K. pneumoniae* sequence type (ST) 307 or *Escherichia coli* ST38) as also its association with MGEs, e.g., OXA-48 linked with different Tn*1999* variants on highly transferable IncL plasmids [10,11]. The expansion of high-risk *K. pneumoniae* clones with a multidrug-resistant (MDR) or extensive drug-resistant (XDR) phenotype has been observed in recent years [4]. *K. pneumoniae* ST147 has been reported as an emerging high-risk clone associated with plasmid-encoded extended-spectrum β-lactamases (ESBLs) like *bla*_CTX-M-15,_ or carbapenemases such as *bla*_OXA-48_ and *bla*_NDM-1_ [4,12,13,14,15,16,17].

In the present study, we characterise the content and genetic structure of MGEs and the clonal relatedness of three OXA-181-producing *K. pneumoniae* ST147 clinical isolates recovered in Germany.

## 2. Results and Discussion

Dissemination of antibiotic resistance is driven by clonal expansion or horizontal gene transfer, including mainly MGEs [1,2]. In the present study, all three isolates colonising haematology/oncology patients were identified as *K. pneumoniae* ST147 and were the only representatives of this ST among 40 in total collected *K. pneumoniae* isolates. The three isolates were also characterised by their capsular type KL64 (*wzi* allele 64). MDR *K. pneumoniae* ST147 isolates represent a successful clone with a global spread and these isolates are often armed with carbapenemases and ESBLs [15,18]. The German National Reference Centre for Multidrug-Resistant Gram-negative Bacteria and the Robert Koch Institute reported, between 2008 and 2014, 13 carbapenemase-producing ST147 *K. pneumoniae* isolates in Germany. In particular, 9/42 OXA-48-, 3/34 KPC-2- and 1/5 NDM-1-producing isolates were assigned to ST147 [19].

In the present study, the isolates HKP0018, HKP0064 and HKP0067 were analysed by whole genome sequencing (WGS) and harboured on the chromosome a gene encoding the intrinsic SHV-11, as well as *oqxAB* and *fosA* genes, belonging to the core genome of the KpI–III phylogroups [20]. The plasmid-encoded resistome of the investigated isolates, summarised in Table 1, was identical and included beta-lactam, aminoglycoside, fluoroquinolone, tetracycline and other antimicrobial resistance determinants. Antimicrobial susceptibility testing showed that all three *K. pneumoniae* isolates exhibited an XDR phenotype; resistant to ampicillin, aztreonam, ceftazidime, chloramphenicol, ciprofloxacin, gentamicin, imipenem, meropenem, minocycline, tetracycline, ticarcillin, tigecycline, and trimethoprim and susceptible only to amikacin and colistin (Table 2). MDR and XDR *K. pneumoniae* isolates involved in nosocomial outbreaks have been widely reported [4,21,22]. Between June and October 2109, an outbreak of XDR *K. pneumoniae* producing NDM-1 and OXA-48 was reported in four medical facilities in Mecklenburg-Western Pomerania, Germany [23]. Molecular characterisation using core genome multi-locus sequence typing (cgMLST) analysis revealed that the three investigated isolates were closely related and formed a cluster with 0–1 allelic differences (data not shown). One could speculate that the closely related isolates were likely transmitted within the hospital. All three patients had been hospitalised in the same department (Table 3) and two of the patients had an overlapping hospitalisation at the same ward (C5A). However, a direct connection to HKP0018 could not be established within the study. 

Phylogenetic analysis of 30 ST147 *K. pneumoniae* isolates from different countries showed several branches (Figure 1). The isolates HKP0018, HKP0064 and HKP0067 were on the same branch with ST147 isolates from different countries, such as Switzerland, USA, United Kingdom and Singapore, illustrating the worldwide spread of this clone. In addition, the three investigated isolates clustered together with 7 ST147 *K. pneumoniae* isolates recovered between 2013 and 2014 in Göttingen, Germany. The latter MDR isolates harboured the carbapenemase OXA-48 on a 63.6 kb IncL plasmid [15]. The close genetic relatedness observed between the German isolates suggests that an OXA-48-like producing ST147 clone is circulating in the country.

In the present study, plasmid analysis revealed eight closed plasmids for each individual *K. pneumoniae* isolate.

OXA-48-like is the most prevalent carbapenem-hydrolysing β-lactamase in Enterobacterales isolates from Germany [9,24]. MDR *K. pneumoniae* ST147 encoding OXA-48 on a conjugative IncL plasmid have been recently reported in Germany [15]. In the present study, all three investigated isolates harboured OXA-181 on an identical 6103 bp ColKP3 plasmid, pHKP0018.1. This plasmid also encoded the mobilisation genes *mobA*, *mobB*, *mobC* and *mobD*, and, upstream of *bla*_OXA-181_ gene, 170 bp of a disrupted IS*Ecp1* was present. A blastn analysis to compare pHKP0018.1 to sequences available in the GenBank database revealed high similarities mainly with three groups of plasmids, of which Carbapenemase OXA-232_ColKP3 (Acc. No CP050165), pKP3-A (Acc. No JN205800) and p50595_OXA_181 (Acc. No CP050375) were chosen as exemplars for a more detailed comparison. The first one, with a size of 6141 bp, showed an identity of 99.98% to our plasmid, has a longer fragment of the interrupted IS*Ecp1* (208 bp) and carries the *bla*_OXA-232_ gene, a *bla*_OXA-181_ variant from which it differs by a single nucleotide, leading to the Arg214-Ser amino acid substitution, and from which it probably originated (Figure 2) [25]. Plasmid pKP3-A, obtained from a clinical *K. pneumoniae* isolate in 2010, is a ColKP3 plasmid carrying *bla*_OXA-181_, proved to be mobilisable but not self-transmissible. It showed 99.95% similarity when compared to pHKP0018.1, from which it differs by the presence of the complete IS*Ecp1* element. In this plasmid, the carbapenemase gene was described as part of the Tn*2013* transposon, made up by the 3139 bp module IS*Ecp1*-*bla*_OXA-181_-Δ*lysR*-Δ*ereA* [26]. In plasmid pHKP0018.1 this transposon was disrupted, with only the two right inverted repeats (IRR1 and IRR2) and the 3′ target site duplication (ATATA) still identifiable (Figure 2) [26]. Lastly, p50595_OXA_181 plasmid depicts the group of X3-ColKP3 plasmids of approximately 51 kb in size, which held 50% of the pHKP0018.1 plasmid, with an identity of 100%. This portion contained the interrupted Tn*2013* (ΔIS*Ecp1*-*bla*_OXA-181_-Δ*lysR*-Δ*ereA*) and an almost complete *repA* gene of ColKP3, inserted between the two insertion sequences IS*3000* and IS*Kpn19* (Figure 2). The sequence comparative analysis also showed that *bla*_OXA-181_ seems to be almost uniquely located on X3-ColKP3 plasmids, frequently harboured by *E. coli* isolates, while its variant *bla*_OXA-232_ is primarily located on ColKP3 plasmids harboured predominantly by *K. pneumoniae*. Nevertheless, both variants are distributed on a global scale, including not only clinical isolates but also animal and environmental ones. Indeed, OXA-181 and OXA-232 represent, respectively, the second and third most common and widespread OXA-48-like enzyme and both are described as part of the Tn*2013* transposon, which, together with its localisation on plasmids like ColE-type, IncX3, IncN1 and IncT, is responsible for their dissemination [10].

S1-PFGE, Southern blot and WGS analysis revealed that all three isolates harboured an IncR plasmid, pHKP0018.2, pHKP0064.2 and pHKP0067.2, presenting only the *repB* gene and lacking the *repE* and *repA* genes and encoding the same antibiotic resistance determinants (Figure 3). The MDR region included a mosaic structure of 12 antibiotic resistance genes, including β-lactamases *bla*_CTX-M-15_ (present in two copies on each IncR plasmid), *bla*_OXA-1_, *bla*_TEM-1B_, aminoglycoside modifying enzymes *aac(6’)Ib-cr*, *aac(3)-IIa*, *strA*, *strB*, as well as the resistance determinants *qnrS1*, *sul1*, *dfrA1*, *tet*(A) and *catB3*-like (Table 1). The MDR region was highly recombinogenic and encoded several copies of different ISs (n = 9). Furthermore, pHKP0018.2, pHKP0064.2 and pHKP0067.2 encoded a *higB*/*higA* toxin-antitoxin (TA) module and *parA/parB* partitioning genes, contributing to plasmid stabilisation and inheritance. As many others previously described, containing only the *repB* gene alone, the IncR plasmid of this study did not harbour known conjugative loci, and consequently attempts to transfer by conjugation IncR and to mobilise the ColKP3-OXA-181 into *E. coli* J53 were not successful. The IncR plasmids showed high sequence homology to IncR plasmids pKp_Goe_304-4 (Acc. No CP018724.1), pKp_Goe_021-4 (Acc. No CP018718.1), pKp_Goe_024-4 (Acc. No CP018705.1), and CP017989.1 of a ST147 *K. pneumoniae* isolate collected in Germany in 2014, and to the IncR plasmid pSg1-NDM (Acc. No CP011839.1) identified in a ST147 *K. pneumoniae* isolate from Singapore [18].

Sequence analysis revealed that pHKP0064.2 and pHKP0067.2 were identical and 70,762 bp in size. Nevertheless, comparative analysis revealed a rearrangement of a composite transposon flanked by two inverted copies of IS*26* and containing *catB3*-like, *aac(6’)Ib-cr* and *bla*_OXA-1_ genes. This 3826 bp region was inserted in the same position in the two IncR plasmids but in opposite orientation. Similarly, another reshuffling of a 13,957 bp region was observed for pHKP0064.2 and pHKP0067.2. This genomic region was flanked by two copies of IS*Ecp1* in inverse orientation and harboured a truncated transposase, Tn3 resolvase, *bla*_TEM-1B_, *qnrS1*, recombinase, IS*Kpn19*, *umuC*, HAMP-domain and IS*26* (Figure 3). In the isolate HKP0018 an IncR plasmid, pHKP0018.2, with a size of 66,330 bp was identified. The plasmids pHKP0064.2 and pHKP0067.2 shared a high degree of sequence homology with pHKP0018.2, apart from a 4432 bp region which was missing from the latter plasmid. The missing region was part of the 13,957 bp genomic region involved in the rearrangement in pHKP0064.2 and pHKP0067.2. This subregion was comprised of genes encoding for the error-prone DNA polymerase V subunit (*umuC*) and a sensor histidine kinase (HAMP-domain) followed by the MGE IS*Kpn19* (Figure 3). These results indicate that within a group of clonal isolates, diversity can still be observed. Genetic variation within clonal bacterial groups caused by homologous recombination has been described in *E. coli* [27]. While genetic rearrangement, such as inversion or duplication, caused by MGEs have been confirmed by diverse studies [28,29,30].

An identical 113,014 bp IncFIB-like plasmid, pHKP0018.3, was identified in the *K. pneumoniae* isolates and did not encode any known antibiotic resistance determinants (Appendix A). However, this plasmid harboured a tellurite/colicin resistance determinant, phage-related genes, and was lacking known conjugative transfer genes. Furthermore, the plasmid harboured two members of the IS3 family, IS*Kpn1* and IS*2*. The IncFIB-like plasmid showed high homology (coverage 97%, identity 100%) to pSG1.1 (Acc. No CP012427.1) from an NDM-1 positive ST147 *K. pneumoniae* isolate from Singapore and also with other ST147 IncFIB plasmids (Acc. No CP021940.1, CP021945.1 and CP014756.1), indicating that this plasmid might be intrinsic to this ST [18].

Southern blot and WGS revealed that the ST147 isolates carried also an identical 54,750 bp plasmid, pHKP0018.4 (Appendix A). The plasmid showed similarity (coverage 78%, identity 99%) to *K. pneumoniae* ST147 plasmids recovered from Singapore, pSg1-3 (Acc. No CP012429) [18]. pHKP0018.4 exhibited also similarity (coverage 62%, identity 83%) to phiKO2 of a *Klebsiella oxytoca* isolate which was described as a prophage able to replicate as linear plasmids with covalently closed ends [32].

Small plasmids, often present in high copy numbers, can serve as an important reservoir for antibiotic resistance determinants, such as small ColE plasmid derivatives encoding *qnrS1* in *Salmonella enterica* [33,34,35]. In the present study, apart from the ColKP3 *bla*_OXA-181_-encoding plasmid, the ST147 *K. pneumoniae* isolates harboured in addition four identical small plasmids, which varied in size from 1.4 kb to 8.4 kb and did not encode known antimicrobial resistance determinants. An identical 8428 bp plasmid, pHKP0018.5, was identified in the three isolates. The plasmid carried two Col-like replication initiation proteins and showed similarity to pKpvST147B_4 (Acc. No CP040727.1, coverage 56%, identity 100%) from a ST147 *K. pneumoniae* isolate recovered at a hospital in south-east England (Appendix A). Another plasmid, 5499 bp in size and identical for the investigated ST147 isolates (pHKP0018.6), was detected and typed as a Col-like plasmid (Appendix A). 

Moreover, a 2044 bp plasmid identical for the three *K. pneumoniae* plasmids, pHKP0018.7, was detected and encoded two hypothetical proteins, with no conserved domains. This plasmid could not be assigned to a replicon type and was identical (coverage 100%, identity 100%) to plasmids p4_1_2.4 (Acc. No CP023843.1) and pDA33140-2 (Acc. No CP029584.1) both from ST147 *K. pneumoniae* isolates recovered in Sweden (plasmid map not shown). Finally, an identical 1459 bp plasmid, pHKP0018.8, replicon typed as Col-like and bearing a hypothetical protein was identified. The Col-like plasmid was identical (coverage 100%, identity 100%) to plasmids found in *E. coli*, such as pEC881_8 (Acc. No CP019021.1) and pEC648_7 (Acc. No CP008721.1), which can be a result of interspecies plasmid transfer (plasmid map not shown).

## 3. Materials and Methods 

### 3.1. Bacterial Isolates and Transformants

The isolates HKP0018, HKP0064 and HKP0067 were recovered in 2015 from throat and rectal swabs of three patients on admission to a university hospital in northern Germany (Table 3). The isolates were collected as part of the CONTAIN multicentre cohort study of the German Centre for Infection Research (DZIF) on the efficiency of infection control measures to prevent the transmission of ESBL producing Enterobacterales in haematology/oncology units [36]. The selection of the three isolates for further investigation was based on their clonal relatedness, ST, and acquired resistome. The surveillance swabs were plated on selective media (chromID^®^ ESBL; bioMérieux, Nürtingen, Germany) and incubated for 18–24 h. The species identification was performed with MALDI-TOF mass spectrometry. Additionally, plasmid DNA was extracted from the isolate HKP0018 with the PureYield Plasmid Midiprep System (Promega, Madison, WI, USA) and then used to transform One Shot MAX Efficiency DH5α-T1R Competent Cells (Thermo Fisher Scientific, Waltham MA, USA). Selection of transformants was performed using ampicillin (40 mg/L) and tetracycline (30 mg/L) and was confirmed by PCR (Supplementary data).

### 3.2. Antimicrobial Susceptibility Testing

MICs for ampicillin, tetracycline, trimethoprim, gentamicin (Sigma–Aldrich, Steinheim, Germany), amikacin, aztreonam, imipenem, meropenem, minocycline, rifampicin, (Molekula, Newcastle-upon-Tyne, UK), levofloxacin (Sanofi Aventis, Frankfurt, Germany), ciprofloxacin (Bayer Pharma AG, Berlin, Germany) and ticarcillin (Carl Roth GmbH, Karlsruhe, Germany) were determined using the agar dilution method [37]. MICs for colistin and tigecycline were determined by broth microdilution method (Merlin Diagnostika GmbH, Bornheim, Germany). *E. coli* ATCC 25922, *Pseudomonas aeruginosa* ATCC 27853, and *Staphylococcus aureus* ATCC 25923 were used as quality control strains. MICs were interpreted using the resistance breakpoints for Enterobacterales from EUCAST (Version 10.0, January 2020, http://www.eucast.org/clinical_breakpoints/) and CLSI (https://clsi.org/standards/products/microbiology/documents/m100/).

### 3.3. S1-Pulsed-Field Gel Electrophoresis (S1-PFGE) and Southern Blot Hybridisation

Plasmid linearisation by S1 nuclease followed by PFGE was used to determine the size and total number of plasmids. Bacterial DNA embedded in agarose plugs was digested using 50 Units S1 nuclease (Thermo Fisher Scientific, Waltham, MA, USA) per plug slice and incubated according to the manufacturer’s instructions. Samples were run on a CHEF-DR II system (Bio-Rad, Munich, Germany) for 17 h at 6 V/cm and 14 °C while initial and final pulses were conducted at 4 and 16 s, respectively. The Lambda PFG Ladder and λ DNA-Mono Cut Mix (New England Biolabs, Frankfurt, Germany) were used as markers. The approximate plasmid size was calculated using Image Lab^TM^ software (Bio-Rad, Munich, Germany).

Southern blot hybridisation was performed to determine the plasmid/chromosomal gene location by hybridisation with digoxigenin (DIG)-labelled probes (Roche, Mannheim, Germany). For the IncR replicon and *strA* of pHKP0018.2 and for the terminase pHKP0018.4 specific probes were used respectively (Appendix A). Signal detection was performed according to the manufacturer’s instructions using CDP-Star® ready-to-use (Roche, Mannheim, Germany) chemiluminescent substrate by autoradiography on a X-ray film (GE Healthcare, Buckinghamshire, United Kingdom). Chromosomal location was shown by colocalisation with a *rpoB* probe.

### 3.4. Whole Genome Sequencing (WGS) and Bioinformatics

Total DNA from the bacterial isolates and transformants was extracted using the MagAttract HMW DNA Kit (Qiagen, Hilden, Germany) and plasmid DNA was extracted using PureYield Plasmid Midiprep System according to manufacturer’s instructions and used for short-read sequencing. Sequencing libraries were prepared using a Nextera XT library prep kit (Illumina GmbH, Munich, Germany) for a 250 bp paired-end sequencing run on an Illumina MiSeq platform. The obtained reads were de novo assembled with the Velvet assembler integrated in the Ridom SeqSphere+ v. 7.2.1 software, and SPAdes 3.11 [38]. Finally, where necessary, overlapping assembly contigs and predicted gaps were filled and confirmed by PCR-based gap closure as described previously [39].

DNA extraction for long-read sequencing was performed using the Genomic-Tips 100/G kit and Genomic DNA Buffers kit (Qiagen, Hilden, Germany) according to the manufacturer’s instructions. Libraries were prepared using the 1D Ligation Sequencing Kit (SQK-LSK108) in combination with Native Barcoding Kit (EXP-NBD103) and Rapid Barcoding Kit (SQK-RBK004) in accordance with the manufacturer’s instructions (Oxford Nanopore Technologies, Oxford, United Kingdom) and were loaded onto a R9.4 flow cell (Oxford Nanopore Technologies, Oxford, United Kingdom). The run was performed on a MinION MK1b device (Oxford Nanopore Technologies, Oxford, United Kingdom). Collection of raw electronic signal data and live base-calling was performed using the MinKNOW software and Albacore (Oxford Nanopore Technologies, Oxford, United Kingdom). *De novo* assembly of the MinION long-reads was performed using Canu [40]. The Illumina short-reads were assembled with the MinION long-reads using hybridSPAdes and Unicycler [41,42]. Additionally, plasmidSPAdes was implemented to identify plasmid sequences [43].

The assembled genomes generated in this project have been deposited in the NCBI under the BioProject ID PRJNA660340 (BioSample accessions: HKP0018, SAMN15946735; HKP00164, SAMN15946736; HKP0067, SAMN15946737).

### 3.5. Molecular Epidemiology, Resistome, Mobilome and Genome Annotation

The Pasteur multi-locus sequence typing (MLST) scheme was used to assign the ST (https://bigsdb.pasteur.fr/index.html). The molecular epidemiology was investigated with a validated cgMLST scheme, including 2358 target alleles, using the Ridom SeqSphere+ v. 7.2.1 software [44]. Capsular type (KL-type) were assigned using Kaptive Web [45]. The resistome and plasmidome were analysed using ResFinder v.3.2.0 (https://cge.cbs.dtu.dk/services/ResFinder/) and PlasmidFinder v.2.0.1 [46,47]. Genome sequences were annotated using the RAST server (http://rast.nmpdr.org/) and partially manually edited. Plasmids were graphically depicted using SnapGene (http://www.snapgene.com/). 

### 3.6. Conjugation Experiments

Broth mate conjugation experiments were performed using the sodium azide-resistant *E. coli* J53 as recipient. Selection of transconjugants was performed using sodium azide (200 mg/L) and ampicillin (40 mg/L), or tetracycline (30 mg/L). Transconjugants were tested by PCR for the presence of the *bla*_OXA-181_ and *tet*(A) genes, while their susceptibility to meropenem (10 μg) and tetracycline (30 μg) was tested using the disk diffusion method, according to EUCAST recommendations (Version 10.0, January 2020, http://www.eucast.org/clinical_breakpoints/).

## 4. Conlusions

In conclusion, the present study describes a complex variety of plasmids within three clonal ST147 *K. pneumoniae* isolates recovered from haematology/oncology patients hospitalised in the same German hospital. The ST147 *K. pneumoniae* isolates harboured the *bla*_OXA-181_ carbapenemase gene on a small ColKP3 plasmid, but also a complex array of 12 antibiotic resistance determinants on an MDR IncR plasmid, severely limiting treatment options. The recombinogenic nature of the MDR IncR plasmid encoding a large number of ISs can serve as genome plasticity mediators. The IncR plasmids of the studied isolates differed overall in a 4 kb region which could be attributed to an IS transposition event, as also in the opposite orientation of two composite transposons (3.8 kb and 13 kb). These results indicate that within a cluster of closely related isolates, variation can be observed due to the dynamic nature of MGEs. The abundant mobilome and resistome of the *K. pneumoniae* isolates combined with the emergence of ST147 as an international high-risk clone has the potential to become a major challenge for the healthcare setting and requires special attention and vigilance.

## Figures and Tables

**Figure 1 antibiotics-09-00675-f001:**
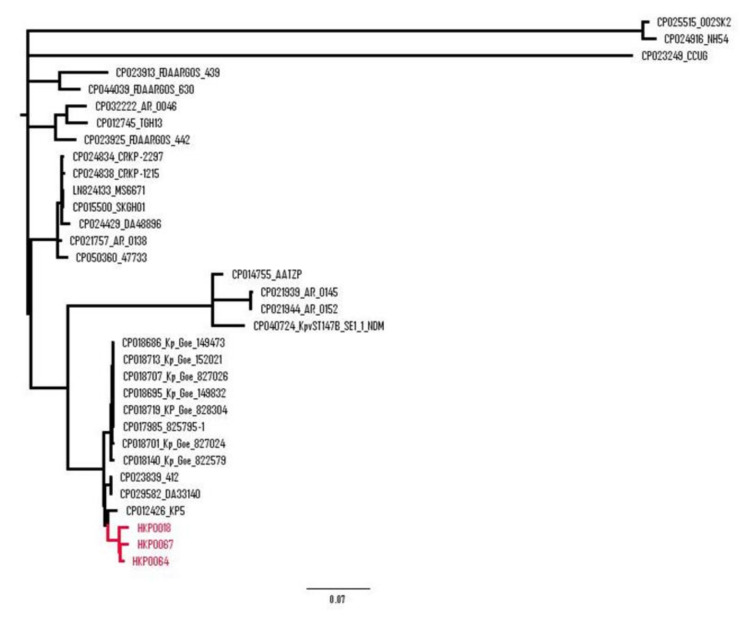
Phylogenetic analysis of HKP0018, HKP0064, HKP0067 and 30 ST147 *K. pneumoniae* isolates. Phylogenetic maximum-likelihood tree was generated using the FigTree v1.4.3 software of the SNP analysis performed using the kSNP3 tool (Galaxy version 3.1) software at the ARIES Galaxy server (https://aries.iss.it/).

**Figure 2 antibiotics-09-00675-f002:**
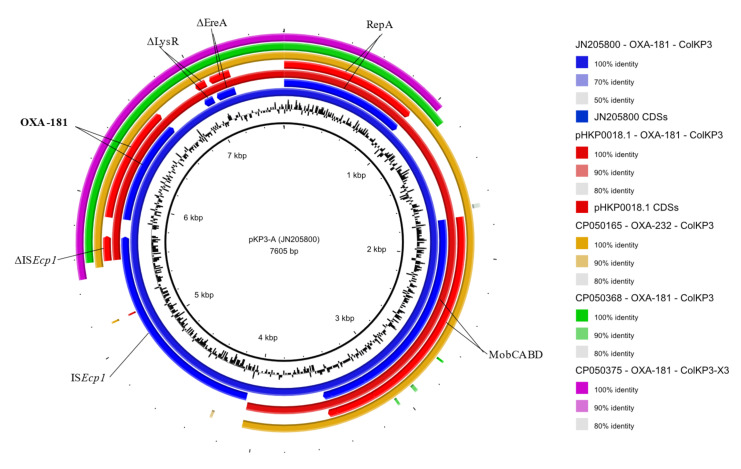
Graphical representation of *bla*_OXA-181_/*bla*_OXA-232_-carrying plasmids sequence comparison. Starting from the inner ring: GC content of pKP3-A plasmid sequenced (here used as reference), *bla*_OXA-181_-positive pKP3-A plasmid sequence (JN205800), pKP3-A CDSs, *bla*_OXA-181_-positive pHKP0018.1 plasmid sequence (CP061063.1), pHKP0018.1 CDSs, *bla*_OXA-232-_positive Carbapenemase (OXA-232)_ColKP3 plasmid sequence (CP050165), *bla*_OXA-181_-positive p47733_OXA_181 plasmid sequence (CP050368), *bla*_OXA-181_-positive p50595_OXA_181 plasmid sequence (CP050375). CDS’s arrows indicate their transcription direction. Hypothetical proteins are not displayed. The figure was generated with BRIG v0.95.

**Figure 3 antibiotics-09-00675-f003:**
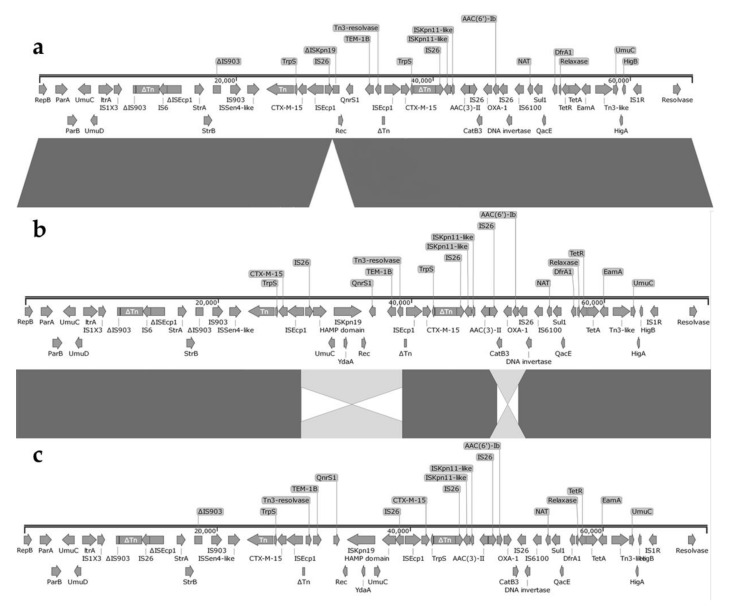
Major structural features of the IncR plasmids, pHKP0018.2 (**a**), pHKP0064.2 (**b**) and pHKP0067.2 (**c**), identified in *K. pneumoniae* isolates HKP0018, HKP0064 and HKP0067, respectively. Arrows indicate the deduced open reading frames (ORFs) and their orientations. Hypothetical proteins are not shown. The figure was generated with EasyFig 2.1 [31].

**Table 1 antibiotics-09-00675-t001:** Plasmid encoded antimicrobial resistance determinants, plasmid content and plasmid size of the isolates.

Plasmid	Replicon	Size (bp)	Antimicrobial Resistance Determinants	Isolate No.		
				HKP0018	HKP0064	HKP0067
pHKP0018.1	ColKP3	6103	blaOXA_-181_	+	+	+
pHKP0018.2	IncR	66,330	*bla*_CTX-M-15_*^b^*, *bla*_OXA-1_, *bla*_TEM-1B_, *aac*(*6’*)*Ib-cr*, *aac*(*3*)-*IIa*, *strA*, *strB*, q*nrS1*, *sul1*, *dfrA1*, *tet*(A), *catB3*-like	+	-	-
pHKP0064.2	IncR	70,762	*bla*_CTX-M-15_*^b^*, *bla*_OXA-1_, *bla*_TEM-1B_, *aac*(*6’*)*Ib*-*cr*, *aac*(*3*)-*IIa*, *strA*, *strB*, *qnrS1*, *sul1*, *dfrA1*, *tet*(*A*), *catB3*-like	-	+	+
pHKP0018.3	IncFIB	113,014	-	+	+	+
pHKP0018.4	NT *^a^*	57,450	-	+	+	+
pHKP0018.5	Col-like	8428	-	+	+	+
pHKP0018.6	Col-like	5499	-	+	+	+
pHKP0018.7	NT *^a^*	2044	-	+	+	+
pHKP0018.8	Col-like	1459	-	+	+	+

*^a^* NT, not typeable; *^b^* gene present in two copies.

**Table 2 antibiotics-09-00675-t002:** Antimicrobial susceptibility of the three *K. pneumoniae* isolates.

Antimicrobial Agent	MIC (mg/L)			Susceptibility *^a^*
	HKP0018	HKP0064	HKP0067	
Amikacin	8	8	8	S
Ampicillin	>128	>128	>128	R
Aztreonam	>128	>128	>128	R
Ceftazidime	128	128	128	R
Chloramphenicol	32	32	32	R
Ciprofloxacin	128	128	128	R
Colistin *^b^*	1	2	1	S
Gentamicin	128	128	128	R
Imipenem	8	8	8	R
Levofloxacin	64	64	64	R
Meropenem	32	32	32	R
Minocycline *^c^*	64	64	64	R
Rifampicin *^d^*	64	64	64	-
Tetracycline *^c^*	128	128	128	R
Ticarcillin	>128	>128	>128	R
Tigecycline *^b^*	2	2	2	R
Trimethoprim	128	128	128	R

*^a^* R, resistant; S, susceptible; *^b^* tested by broth microdilution method; *^c^* only CLSI breakpoint available; *^d^* no breakpoint available.

**Table 3 antibiotics-09-00675-t003:** *K. pneumoniae* clinical isolates information.

Isolate	Date of Isolation	Source	Department	Ward	ST
HKP0018	16.02.2015	Rectal swab	Haematology/Oncology	C5A	147
HKP0064	08.05.2015	Throat swab	Haematology/Oncology	1G	147
HKP0067	19.05.2015	Rectal swab	Haematology/Oncology	C5A	147

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
