# Peer review of "Antibiotic Resistance and Mobile Genetic Elements in Extensively Drug-Resistant Klebsiella pneumoniae Sequence Type 147 Recovered from Germany"

_antibiotics, 2020, doi:10.3390/antibiotics9100675_

Round 1

Reviewer 1 Report

The Authors investigated the antibiotic-resistance and mobile genetic elements in extensively drug-resistant Klebsiella pneumoniae sequence type 147 recovered from Germany. The work is well organized and clearly written. References are also pertinent. I suggest accepting the manuscript with minor revisions.

In particular:

- Introduction (lines 46-67): the introduction is too short and must be extended.

- Table 2 (line 99): the Authors should report EUCAST and CLSI breakpoints (µg/ml) and interpretive criteria (S, I, R) for Klebsiella pneumoniae.

- Figure 2 (line 151): image resolution must be enhanced.

Author Response

- Reviewer 1: Introduction (lines 46-67): the introduction is too short and must be extended.

- Authors: Thank you for this suggestion. The introduction has been extended.

- Reviewer 1: Table 2 (line 99): the Authors should report EUCAST and CLSI breakpoints (µg/ml) and interpretive criteria (S, I, R) for Klebsiella pneumoniae.

- Authors: The interpretive criteria (S and R) have been added to Table 2 according to the reviewer’s suggestion. Regarding the EUCAST and CLSI breakpoints we cite in the methods (see lines 282-285) the version of the breakpoints table used in the present study.

- Reviewer 1: Figure 2 (line 151): image resolution must be enhanced.

- Authors: The resolution of Figure 2 has been improved.

Reviewer 2 Report

Xanthopoulou K et al. “Antibiotic resistance and mobile genetic elements in extensively drug-resistant Klebsiella pneumoniae sequence type 147 recovered from Germany” (Antibiotics-953036)

The authors analyzed three XDR K. pneumoniae isolates, focusing on plasmids.

Introduction. The authors mentioned that this study focuses on mobilome in antibiotic resistant K. pneumoniae isolates. Thus, more information on mobilome should be included.

The authors analyzed three isolates. Probably, more XDR K. pneumoniae isolates could be obtained. Please represent the XDR rate in surveillance study. In addition, it is need to explain what criteria these three isolates were selected and analyzed.

The authors performed WGS for three isolates. Probably, whole genome data could be obtained. However, they don’t mention the data of chromosome. Do the three K. pneumoniae isolates showed the same chromosome structure and sequence?

As I think, Figure 1 is not required in this paper.

Line 235-237. The authors analyzed only three isolates from Germany. This conclusion is too enlarged interpretation. Probably, other plasmid sequences of K. pneumoniae ST147 or closely related clones can be obtained. If the authors analyzed them together, it will help to support the conclusion of this paper.

Author Response

Xanthopoulou K et al. “Antibiotic resistance and mobile genetic elements in extensively drug-resistant Klebsiella pneumoniae sequence type 147 recovered from Germany” (Antibiotics-953036)

 The authors analyzed three XDR K. pneumoniae isolates, focusing on plasmids.

- Reviewer 2: Introduction. The authors mentioned that this study focuses on mobilome in antibiotic resistant K. pneumoniae isolates. Thus, more information on mobilome should be included.

- Authors: The introduction has been extended and more information about mobile genetic elements has been included.

 - Reviewer 2: The authors analyzed three isolates. Probably, more XDR K. pneumoniae isolates could be obtained. Please represent the XDR rate in surveillance study. In addition, it is need to explain what criteria these three isolates were selected and analyzed.

- Authors: We thank the reviewer for this helpful comment. The three isolates were further investigated because they were the only ones to form a cluster and were also the only ST147 isolates from the surveillance study. ST147 is considered as a high-risk clone (HRC) which is why we concentrated on it. We have included a sentence (see line 81-82) about the number of isolates collected in total (a publication is planned for the study). We have also added a sentence about how the isolates were selected according to the reviewer’s suggestion (see line 263-264). With regards to the rates of XDR, this is work underway and is not finished. We would not like to misrepresent the XDR rate by publishing Vitek results.

 - Reviewer 2: The authors performed WGS for three isolates. Probably, whole genome data could be obtained. However, they don’t mention the data of chromosome. Do the three K. pneumoniae isolates showed the same chromosome structure and sequence?

- Authors: In lines 100-102 we state that: “Molecular characterization using cgMLST analysis revealed that the three investigated isolates were closely related and formed a cluster with 0-2 allelic differences (data not shown).” referring to the close genetic relationship and clonality of the investigated isolates. We have also described that the isolates also harboured identical chromosomally encoded antibiotic resistance genes (lines 89-91). Furthermore, the close genetic relatedness of the three isolates is also shown in the phylogenetic maximum-likelihood tree (Figure 1). We think that there is no further need to elucidate chromosome encoded structures.

- Reviewer 2: As I think, Figure 1 is not required in this paper.

- Authors: I disagree, this figure shows the relatedness of the three isolates amongst themselves (which also covers the previous comment from the reviewer) as well as with other ST147 isolates in Europe. Furthermore this figure illustrates that the ST147 clone is circulating in Germany (see genetic relatedness with the with 7 ST147 K. pneumoniae isolates recovered in Göttingen, Germany).

- Reviewer 2: Line 235-237. The authors analyzed only three isolates from Germany. This conclusion is too enlarged interpretation. Probably, other plasmid sequences of K. pneumoniae ST147 or closely related clones can be obtained. If the authors analyzed them together, it will help to support the conclusion of this paper.

- Authors: We have investigated the only three ST147 isolates that we found. However, as we can see from Figure 1, these isolates are closely related to other ones from Germany, furthermore, ST147 is recognised as a HRC. Our three isolates had 8 shared plasmids and 13 plasmid encoded antibiotic resistant determinants, and were in addition XDR. Therefore, putting all these data together leads one to the conclusion that there is a need for vigilance and this HRC has potential to spread further and be difficult to treat. To support this finding, we have included the manuscript published by Peirano et al., in September 2020 in AAC about the emerging antimicrobial-resistant HRC K. pneumoniae ST147.

Round 2

Reviewer 2 Report

The authors responded appropriately to all of my questions.